

# A novel IoT-based health and tactical analysis model with fog computing

Aykut Karakaya[1] and Sedat Akleylek[2]

[1] Department of Computer Technologies, Bulent Ecevit University, Zonguldak, Turkey
[2] Department of Computer Engineering, Ondokuz Mayis University, Samsun, Turkey

## ABSTRACT

In sports competitions, depending on the conditions such as excitement, stress, fatigue, etc. during the match, negative situations such as disability or loss of life may occur for players and spectators. Therefore, it is extremely important to constantly check their health. In addition, some strategic analyzes are made during the match. According to the results of these analyzes, the technical team affects the course of the match. Effects can have positive and sometimes negative results. In this article, fog computing and an Internet of Things (IoT) based architecture are proposed to produce new technical strategies and to avoid disabilities. Players and spectators are monitored with sensors such as blood pressure, body temperature, heart rate, location etc. The data obtained from the sensors are processed in the fog layer and the resulting information is sent to the devices of the technical team and club doctors. In the architecture based on fog computing and IoT, priority processes are computed with low latency. For this, a task management algorithm based on priority queue and list of fog nodes is modified in the fog layer. Authentication and data confidentiality are provided with the Federated Lightweight Authentication of Things (FLAT) method used in the proposed model. In addition, using the Software Defined Network controller based on blockchain technology ensures data integrity.

## INTRODUCTION

With the use of technological developments in the fields of health and sports, more efficient methods are obtained than traditional ones. Data such as player health, spectator health, and tactical analysis in the match are very important. Instantly checking the health status of people on and off the field, and early detection of certain events in tactical analysis make the competition better. For this, applications based on the Internet of Things (IoT) are quite suitable. Most IoT applications use cloud services for storage and computing. Using fog computing improves efficiency for a large-scale IoT application involving spectators, players and other actors. Since a sports field is within certain limits, the data can be processed on local servers instead of the remote cloud server. This ensures that responses are delivered to the user with low latency because it is important to get quick response to technical team and doctors in terms of early intervention (*Ikram, Alshehri & Hussain, 2015*).

Corresponding author
Aykut Karakaya,
aykut.karakaya@bil.omu.edu.tr

Fog computing offers decentralized architecture in cloud computing by expanding storage, computing and network resources to the edge of the network to support large-scale IoT applications. Cloud and fog provide computing, storage, application, infrastructure and data sources (*Luan et al., 2015*). There are some important differences between them. The main difference is accessibility and proximity. The fog is close to the nodes in the system and is usually located on the local network. Cloud, on the other hand, is the server or data center accessed anywhere on the internet. Fog extends the cloud using virtualization to create virtual sensors and networks. In other words, fog works like a layer between nodes and the cloud (*Mukherjee et al., 2017*). It helps with data analysis, data processing and filtering and increases security for sensitive data. While the cloud is more central, the fog works more efficiently in distributed applications. Important points in terms of tactical analysis in sports: determining the most appropriate rust analysis, helping coaches in replacement, taking measures against problems that may arise when the tactical order is broken, etc. Important point in terms of health is determining the negative health conditions of players and spectators during the match, such as disability.

There are studies in the literature for tactical analysis. With Voronoi graph and Delaunay triangulation, possible pass combinations and analysis of quality pass conditions were created (*Horton et al., 2015*). Detailed data such as how fast, strong, smart for a player was collected and could be analyzed during the match (*Burn-Murdoch, 2018*). Some well-known methods are: the shading method that predicts the most probable moves of the players against certain events, the dominant regions method that enables determining the regions where a player can arrive earlier than other players using the Voronoi graph (*Taki & Hasegawa, 2000*).

## Related Work

In a study conducted as a feedback system in sports, data were expanded to examine exercise practices (*Baca & Kornfeind, 2006*). Trials were made in table tennis, biathlon and rowing sports. In table tennis, the system detects the effects of the ball on the table in order to determine the accuracy of the shot. In the biathlon, the position of the rifle barrel before and after the shot was analyzed with a laser positioning system. In rowing, the system has been calculated the effort made for rowing. In this study, the system is not flexible, as it depends on sports. In another study, the health status of marathon runners was monitored with a WSN (*Pfisterer et al., 2006*). Data was collected with sensors in runners and analyzed offline in a database. Systems that provide video feedback for athletes could also compare athletes (*Vales-Alonso et al., 2010*). However, these studies are not real time.

In real-time studies, the perceived data with a WSN monitoring cyclists were compared with predefined datasets and provides feedback to the group of cyclists. The system could tell the cycling group to change their order, divide or increase their speed. Environmental data are not taken into consideration in this study. However, environmental factors such as wind speed, temperature and humidity can also affect this situation. In a dynamic study dealing with environmental factors, spline-based numerical

approach techniques were recommended as a decision making method (*Vales-Alonso et al., 2010*). Besides the physical conditions of the athlete, the conditions of the land were also handled and the athlete was guided. The spline approach has been used because some structures such as Support Vector Machine (SVM) previously required a lot of data. In this way, multivariate problems could be solved. However, this study has been suggested for training situations in sports such as running and cycling. This method is used for health monitoring and determining the direction of the athlete. The proposed model, besides monitoring the health of players in football, helps the technical team by performing tactical analysis during the match.

In *Kubler et al. (2017)*, it was shown that the emergency situations obtained from the data in the stadium were transmitted to the health institutions in smart city planning. Similarly, it was mentioned that structures that are safe and can detect attacks can be developed. It was emphasized to offer a better sporting event in the 2022 World Cup. *Kubler et al. (2017)* presents a framework that enables IoT service stakeholders to freely join, contribute, and benefit from an open IoT ecosystem. In *Naha et al. (2018)*, it was mentioned that fog computing based structures will be used a lot in the future and real time application architectures should be developed. It was emphasized that security mechanisms are also very important in the proposed models. In addition, resource allocation is a critical issue in distributed resources such as fog. It was explained that the simultaneous processes should be planned appropriately in the fog layer. In *Habibi et al. (2020)*, it was emphasized that architectural design, algorithms and technologies are the top three research subjects. Each of these three aspects can be applied in a number of subject areas that are divided into six major categories: computing paradigm, application, software, networking, computing resource management and security. In *Niu et al. (2019)*, a task assignment algorithm was proposed for edge calculation. It was emphasized that complexity is difficult to calculate in heuristic approaches, so these approaches should be developed. In *Iqbal et al. (2020)*, a task offloading algorithm was proposed. It was explained that the proposed framework can be improved by adding verification mechanisms with blockchain structures. In *Wazid et al. (2020)*, the importance of using blockchain-based cloud or fog for a safe health system was emphasized.

There are also health monitoring studies applied in hospitals, sports fields or other fields. In *Ikram, Alshehri & Hussain (2015)*, the system that detects conditions such as disability of players and sends them to the coach was recommended. Machine to Machine (M2M) standard was used in this system. Also, there were three layers: the detection layer consisting of Wireless Body Area Networks (WBAN), the network layer using ZigBee technology and the cloud layer using cloud services to control all data. Another health monitoring system that uses fog computing was recommended for hospitals in *Paul et al. (2018)*. In this model, data detected using Wireless Sensor Network (WSN) and WBAN were transmitted to the fog layer. In the fog layer, the data was prioritized and the tasks are distributed appropriately to the fog nodes. Thus, more important transactions were given priority. Fog computing has some security advantages. However, security operations between the fog and the cloud were provided using the key pair. It was emphasized that the model is partially implemented due to obstacles such as non-existent

software and complexity of existing software (*Paul et al., 2018*). In *Soni, Pal & Islam (2019)*, a three-factor mutual authentication scheme was proposed in WSNs for the security of health systems. It provided mutual authentication between a user and the sensor node that is a trusted authority.

Software Defined Network (SDN) and Network Function Virtualization (NFV) based structures were used to establish, secure, scale, increase efficiency and reduce the cost of the fog network (*Krishnan, Duttagupta & Achuthan, 2020*). SDN covered software-based technologies, models and protocols that make the global network smarter, more abstract and functional. NFV included high-volume structures developed to virtualize network functions and increase network integration. The task of the fog network was to connect each component of the fog layer. It was difficult to manage these networks and achieve sustainable connectivity. SDN and NFV techniques were used for operations that increase the efficiency of the system such as easy management and scalability (*Yi, Li & Li, 2015*). Each fog node could communicate multi-hop (node-to-node). Therefore, each node served as a router in the fog network. SDN was used for efficient routing optimization in the fog platform. Hierarchical SDN-based fog architecture studies were carried out on the IoT platform in order to enable effective routing of fog nodes (*Okay & Ozdemir, 2018*). NFV separated software from hardware with virtualization to make network functions faster. NFV-based hybrid (cloud and fog) systems were developed to minimize the cost of the network (*Mouradian et al., 2019*).

In *Madsen et al. (2013)*, important measurements were made for the service quality of the fog network. These were connectivity, reliability, capacity and latency measurements. The network was expected to be fast and reliable in connectivity measurement. In reliability measurement, it was expected that the data will be transmitted accurately and completely. However, for this, some control operations were applied to the data. This brought additional costs to the system as a delay. In capacity measurement, the network was expected to use bandwidth and storage areas efficiently. Therefore, the data was collected, then the computing was made. Thus, the cache could be redesigned for other processes (*Wang et al., 2014*). This process caused delay. In the delay measurement, the nodes were expected to receive a rapid response from the computing and storage processes. With the techniques such as flow mining and complex event processing, delays could be reduced by predicting future operations. Interface and programing models were required for IoT developers to move their applications to the fog computing platform. Dynamic, hierarchical resources were needed to optimize based on these models for components to adapt to these platforms (*Yi, Li & Li, 2015*). End nodes could be mobile in IoT applications. Mobile end nodes created challenges in resource management because bandwidth, storage, computing, delay situations change dynamically. Resource management and sharing studies were carried out that properly share heterogeneous resources such as CPU, bandwidth, storage, in the fog layer (*Liu et al., 2014*). The task scheduling model was recommended for IoT applications in the cloud (*Basu et al., 2018*). A hybrid system has been designed by using the genetic algorithm and the ant colony algorithm together. It was emphasized that the performance increased depending on the

number of processors. It was aimed to reduce the processing time of the tasks in the processors rather than the assignment of the tasks to the processors.

In *Kuang et al. (2019)*, a two-level alternation method framework based on Lagrangian dual decomposition was proposed at the lower and upper levels to solve the problems of offloading and resource allocation in MEC systems. In *Zhu et al. (2018)*, a deadline sensitive MEC system with mobile devices sharing multiple heterogeneous MEC servers and formulating a minimum energy consumption problem was proposed. In *Alameddine et al. (2019)*, a new thoughtful decomposition based on the Logic Based Benders Decomposition technique was designed for the resource allocation problem. In *Rahbari & Nickray (2019)*, the importance of resource allocation and task scheduling in fog-based applications was emphasized. A greedy knapsack-based scheduling algorithm has been proposed to properly allocate resources to modules in the fog network. Algorithm simulated with iFogSim. In *Heydari, Rahbari & Nickray (2019)*, it was mentioned that resource management is an NP-hard problem to reduce energy consumption in the cloud. Fog scheduling structures using the Bayesian classification scheduling method were examined. The method was simulated using the iFogSim program and was said to reduce energy consumption and costs in the cloud. In *Fang et al. (2018)*, time sharing optimization of communication and computing resources was discussed in order to minimize the total energy consumption of mobile devices in a Mobile Edge Computing (MEC) system.An algorithm based on the alternating direction method of multipliers was proposed for energy consumption in MEC systems. It was emphasized that these models are simulated and their performance is high.

## Motivation and Contribution

In this article, secure IoT model is proposed to monitor the health of players and spectators, and tactical analysis in the match. In the proposed model, the system becomes more efficient with the help of fog computing. In addition to some security situations of fog computing, there are important features to be achieved such as privacy, confidentiality, authentication, detection of attacks and bandwidth saving, which makes the system safe and fast. Thus, their health is monitored by protecting the privacy of players and spectators from third parties. In tactical analysis, two teams are isolated from each other by authentication methods. In the transmission of both requests/ responses sent to end devices, fog computing is faster than cloud computing. Also, important processes are processed faster and the delay is effectively reduced with the priority based queue method in fog nodes. Thus, the requests with high priority are processed with low latency. In processes that require high processing capacity such as authentication and privacy in the proposed system, bandwidth is saved thanks to implicit certificates. Malicious changes are noticed with block chain based SDN controller. Data loss is prevented and data integrity is ensured during flooding attack. Data stored permanently is encrypted when it is sent from the fog nodes to the cloud. In this article, security problems faced by fog computing systems are also discussed. The measures of our model against these problems are explained. There is also an analysis of our model in terms of safety and effectiveness.

The difference from the similar work in *Ikram, Alshehri & Hussain (2015)* and *Soni, Pal & Islam (2019)* is that the fog model is used in the proposed model. Thus, responses with low latency are produced. A task management algorithm different from the work in *Paul et al. (2018)* is proposed in fog nodes. In the proposed model based on priority queue and node list, it is aimed to make priority processes with low latency. Also, another difference of the proposed model from these studies is the processing of tactical analysis cases as well as the health system. In *Horton et al. (2015)* and *Taki & Hasegawa (2000)* included tactical analysis approaches or reviews. No studies on health systems have been conducted.

In *Baca & Kornfeind (2006)* and *Pfisterer et al. (2006)* were non real time models. In these studies, fog computing was not used and task management mechanism was not recommended. There is no detailed security proof. The proposed model is real-time and includes blockchain-based and lightweight plugins for security principles such as authentication, privacy, integrity. Analysis of the model is also included in this article. In addition, in the proposed model, light protocols are used for communication and security. And so the processing load on low power devices is reduced.

## Organization

In this article, fog computing and IoT based lightweight and safe model are recommended for tactical analysis and health monitoring in sports. In addition, research is made on the features and security principles of the fog computing platform. The following sections of the article include: Features and security needs of fog computing in "Proposed Model", the details of the proposed model in "Analysis of the Proposed Model", the security and effectiveness analysis of the proposed model in "Simulation of the Proposed Model", challenges of wearable technology studies and suggestions for future studies in "Challenges in Wearable Technologies and Future Works" and conclusion in the last section.

## FOG COMPUTING FEATURES AND SECURITY NEEDS

This section contains features and security needs of fog computing. It includes the security concept of fog computing and IoT protocols used in the proposed model. Fog computing is a localized and virtualized platform that provides services such as data processing and storage between end devices and cloud data centers (*Bonomi et al., 2012*). IoT becomes more efficient and more secure with fog computing (*Abdulkareem et al., 2019*). Sensors with restricted resources are used in the IoT applications. Processes that require large computing such as data processing, storage, and encryption are not usually done by sensors. Therefore, there must be a structure that performs these processes. Generally, cloud nodes serving as remote servers or fog nodes serving locally are used.

Fog computing is used to process, store and protect data from sensors without being sent to the cloud. The unsecured Internet network is used when sending data to the cloud. If IoT devices send the detected data to the cloud, it carries risks such as stealing and modifying the data. In addition, it takes a lot of time to process and store the data. The delay time increases for the response to end devices. Since data is processed locally when using fog computing, latency is reduced for response to end devices. Also, if the data

needs to be sent to the cloud, the data is protected with encryption. It is important to use fog computing in IoT applications to increase the performance of the system. A large amount of data is obtained in IoT applications. To transfer the data to the cloud, it requires high bandwidth. Therefore, analyzing the data in the fog layer and sending only the data to be stored permanently to the cloud reduces the bandwidth (*Cisco, 2015*).

Three-layer architecture is commonly used for fog computing. The first layer consists of end devices such as sensor nodes, smart devices, IoT supported devices. The second layer consists of fog nodes such as router, gateway, switch, access points. The fog nodes carry out storage and information processing activities. The third layer consists of remote cloud servers. It provides sufficient storage and information processing services (*Mukherjee et al., 2017*). The data obtained from the end devices (sensors) having the restricted source in the first layer are processed by the fog nodes in the second layer. Then, this data is sent either to the cloud in the third layer or to other end devices (smartphone, etc.) in the first layer. Thus, latency is reduced and data to the cloud is preserved.

Important advantages of fog computing (*Bonomi et al., 2012*): Low latency, preprocessing of data to the cloud for permanent storage, and so saving bandwidth, data processing without connecting to the Internet, compatible with many sensor nodes, real-time applications can be produced, and sufficient resources for data processing and calculation. Due to these advantages, fog computing is used in the tactical analysis and health monitoring model we proposed.

## Security requirements for fog computing in IoT applications

Although fog computing provides some security capabilities to IoT systems, it has security and privacy needs against potential threats. The data detected in the end nodes must be protected when transferring between devices. Fog computing is an important interlayer between the cloud and the user to reduce latency and protect data. However, the fog is exposed to some threats due to wireless transmission. There are security principles and protocols for this.

### Security concepts of fog computing

The security needs in fog computing and the methods used for these needs in the proposed model can be listed as follows (*Mukherjee et al., 2017*; *Alrawais et al., 2017*; *Roman, Lopez & Mambo, 2018*):

- Authentication: Resource-restricted IoT devices cannot perform the encryption required for authentication. For this, it provides the costly storage and data processing needs from outside sources such as fog nodes. Users and nodes authenticate in the fog network to get service. Authentication is provided in the proposed model with the FLAT method.
- Confidentiality: Data must be secured during transmission from the IoT node to the fog node or from the fog node to the cloud. For secure communication, the IoT device communicates with any fog node in the fog network when it needs data processing and storage. Since the resources of the end nodes are restricted, lightweight structures are

used to secure this communication. Therefore, FLAT method is preferred in the proposed model. In the FLAT model, fog nodes make all processes that require high computation for security.

- Malicious and Unauthorized Nodes Prevention: A malicious node in the IoT environment leads to the capture, misdirection and replacement of data. When devices on the network cannot mutually verify each other, the attacker can initiate attacks such as Rogue Gateway, DoS (Denial of Service) and DDoS (Distributed DoS) by continuously sending requests to the fog node. Access to the nodes is limited to protect them from malicious nodes. In the proposed model, authentication is provided by FLAT method. In addition, data integrity is checked with blockchain based SDN structure.

- Data Integrity: In attacks such as Man-In-The-Middle (MITM), which can occur in fog nodes, data corruption must be perceived by fog because the data must be transmitted to the target node without any disruption. If not transmitted, the fog nodes must perform the predefined processes. In the proposed model, blockchain based SDN controller method is proposed for data integrity.

- Accessibility: It defines that users can always get service from sources of fog nodes. DoS and DDoS are some of the attacks that prevent accessibility. These attacks should be prevented.

- Heterogeneity: Data obtained from a large number of devices with different characteristics must be transmitted to the fog nodes. Computing and data processing costs increase due to different communication needs. In the proposed model, devices with different features can communicate wirelessly over the ZigBee network because they support 802.15.x protocols.

- Computing Cost: Fog interlayer is used to reduce latency in IoT applications. The fog layer needs computing capacity for processing and storing data, generating real-time responses, encryption, etc. These tasks are difficult to perform on resource-constrained devices. In the proposed model, high computing processes are made in fog nodes. Thus, the accessibility of the system is maintained.

### Overview of IoT protocols

In the IoT systems, different lightweight protocols are used together with the TCP/IP protocol set. These protocols provide effective communication for resource-constrained structures. In this section discusses the lightweight protocols used in the proposed model.

- The Constrained Application Protocol (CoAP): It is a web transport protocol that is specialized for use in low power and loss networks with restricted nodes (*Shelby, Hartke & Bormann, 2014*). It is defined in RFC 7252. It can work with low power and limited devices up to 10 KiB data size and 100 KiB code size (*Bormann, Ersue & Keranen, 2014*). CoAP is implemented on UDP by default for minimum resource usage. CoAP has a header structure of 4 Bytes (optionally 4*4 B) (*Shelby, Hartke & Bormann, 2014*). CoAP consists of 4 layers. It provides low power multicast support.

It can easily interact with HTTP. In the proposed model, CoAP is used in the communication management of fog nodes and end devices.

- ZigBee protocol: ZigBee is an open and global standard used for wireless personal area networks in low power applications. It can transmit 250 Kbit/s at distances up to 100 meters indoors and over 300 meters outdoors. ZigBee uses the AES-128 structure in its standard security mechanism (*Li, Jia & Xue, 2010*). In the proposed model, the data obtained from the sensors are transmitted by ZigBee networks due to reasons such as the width of the coverage area in the outdoor, low power and supporting a large number of IoT nodes.

## PROPOSED MODEL

This section contains overview of the proposed model. It includes task management in the fog nodes, security details. An IoT model is proposed on health and tactical analysis monitoring in sports using fog computing. Appropriate sensors are placed as a wearable technology for players and spectators. Thanks to the data obtained from these sensors, health conditions are monitored and team-based tactical analysis is performed. In addition, environmental factors (temperature, humidity, ball position etc.) are collected from the sensors. The data obtained from the sensors are transmitted locally and wirelessly to the fog nodes in the ZigBee network. The data is processed in fog nodes and results are sent to the club doctor and the technical team. Since there is a local network, data is transmitted more safely and quickly. The data flow and ZigBee structure of the proposed model are shown in Fig. 1.

### Details of the proposed model

In the proposed model, data is collected from players or spectators in the sports competition. Since the received health data is huge and responses with low latency are required, processing and storing of the data should be done quickly. Also, the data obtained from the ball and players are processed with low latency and position analysis should be done. For this reason, fog computing is used in the proposed model. Also, if there is data to be stored permanently in the cloud system, the data can be encrypted when it is sent to the cloud via the internet. Sensors such as body temperature, pulse, distance and motion, sensitive position, ECG are placed in players. These sensors are found on the player's clothing or directly on their body, such as wristbands and body bands. In this way, WBAN are created. In addition, health problems that require first aid that can occur in the spectator are monitored. Therefore, it is assumed that this WBANs are designed for the spectator as well as the players. These products are difficult to provide as wearable technology is still developing. Therefore, wearable technologies are not included in the scope of this article. In this article, a safe fog-based infrastructure is proposed for health and tactical analysis in the field of sports.

Data flow in Fig. 2: the data obtained from players or spectators with WBAN are transmitted to fog servers; then the results of the data processed in the fog servers are transmitted to the devices of the technical team and club doctors. The local operation of

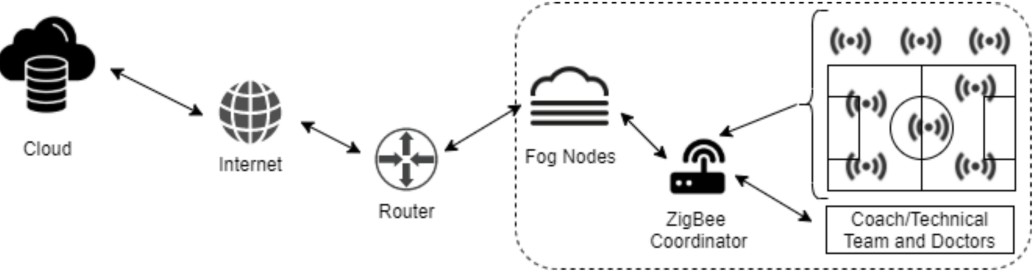

**Figure 1 The data flow and ZigBee structure of the proposed model.**

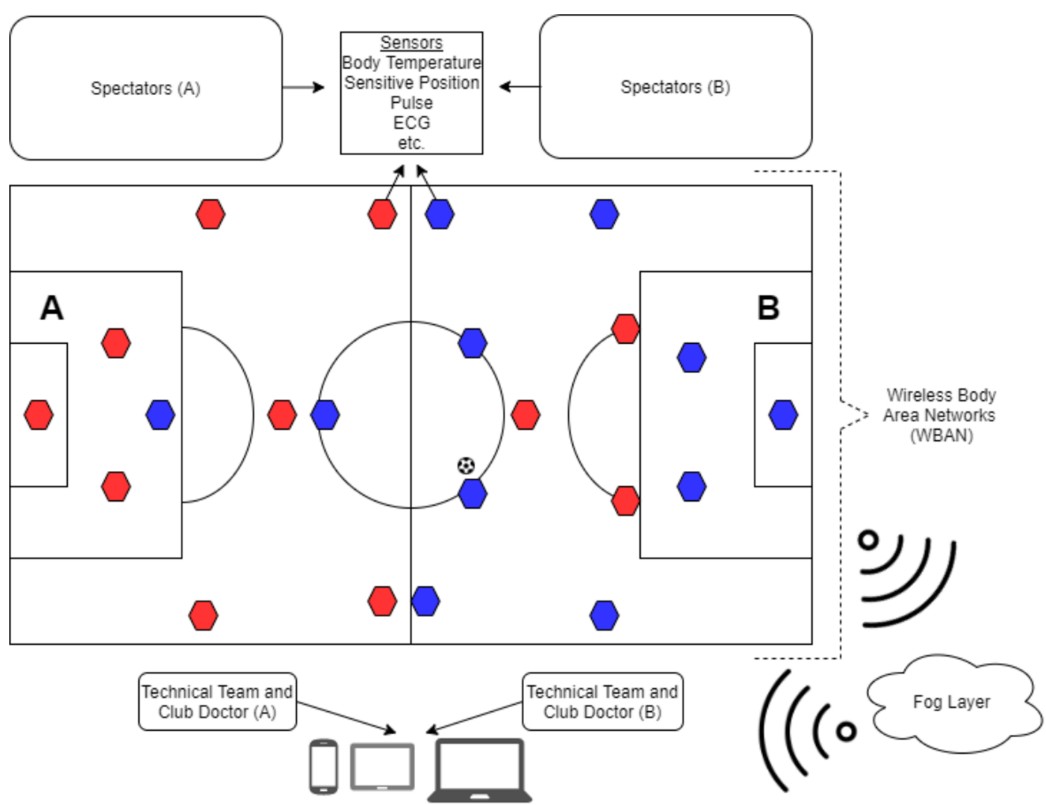

**Figure 2 Sensors and their possible locations in system actors.**

the system reduces latency in data processing and transmission. This helps early treatment if players or spectators are injured. In addition, according to the analysis, it provides the technical team to apply tactical changes early. In order to protect the position and tactical data between the two teams, the nodes of each team must authenticate. Data communication of end node devices performing sensing and monitoring is shown in Fig. 3.

Federated Lightweight Authentication of Things (FLAT) is used for secure data communication and nodes to recognize each other. In security operations, bandwidth is reduced by using Implicit Certificates. ZigBee network is used to transmit the data

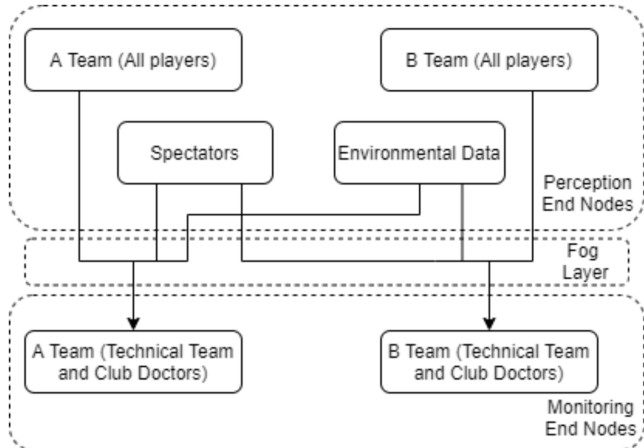

**Figure 3 Data communication of end node devices performing sensing and monitoring.**

obtained from the sensors to the fog nodes. The ZigBee network has its own security methods. In addition, block chain based SDN controller mechanism is used for recording transactions and checking data integrity. Traditionally, in IoT applications that use only cloud, data is sent either without encryption or with encryption on resource-restricted end nodes. This causes either the data to be stolen or the end device to use its resources inefficiently. In the proposed model, security and efficient usage of resources are provided since the detected data is encrypted in the fog nodes and sent to the cloud. In addition, all operations with high computing power in the security phase are performed in the fog nodes. Therefore, the proposed model is lightweight and efficient.

The layers of the proposed model are given in Fig. 4.

- The perception and end node layer consists of sensors used for collecting data and user end devices.
- In the middle layer, a network is required to transmit the data to the fog layer. This network is local and uses the ZigBee protocol, which allows large amount of devices to communicate.
- In the fog layer, data processing and storage are performed by fog nodes. After the data is processed, it is sent to the perception and end node layer or to the cloud layer for permanent storage. Data sent to the cloud layer is encrypted in the fog layer due to the insecure internet.
- The network layer includes the internet connection between the fog and the cloud layer. This layer exchanges data between the cloud and fog layers.
- The cloud layer carries out the permanent storage and processing of data.

The data obtained from the sensor and end node layer are sent to the fog nodes in the fog layer via ZigBee networks in the middle layer. It is processed in fog nodes after passing through task management processes. There are two cases after the results are obtained. The first case is that the results are sent to the cloud layer via the internet in the network

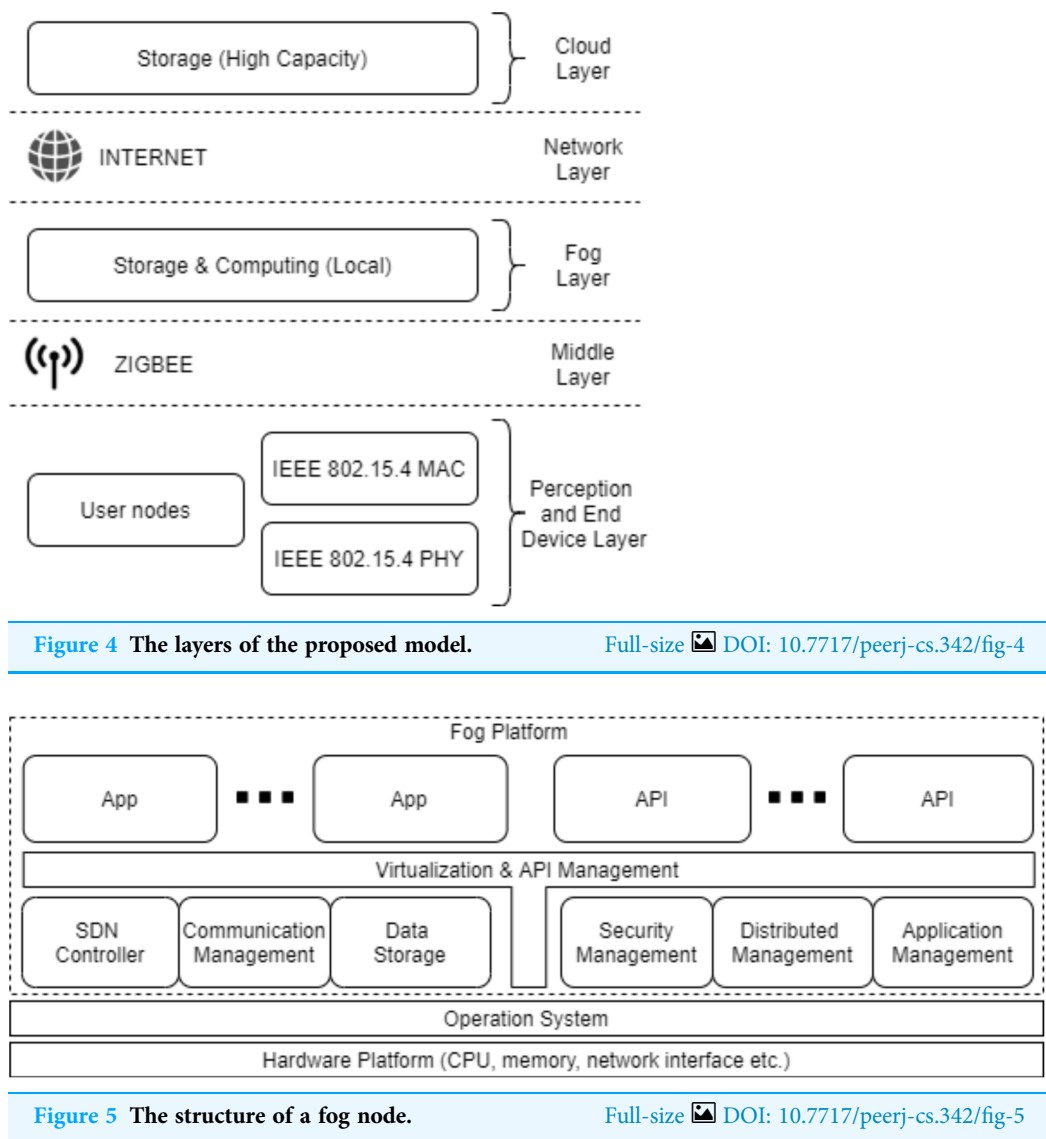

**Figure 4 The layers of the proposed model.**

**Figure 5 The structure of a fog node.**

layer for permanent storage. The second case is that the results are sent to the sensor and end node layer user devices via ZigBee in the middle layer to inform system actors. Devices in the sensor and end node layer are in the same layer as they are located in the same area. Sensors collect data, while end devices report the result to system actors.

## Task management in the fog nodes

Fog nodes perform data processing and storage within the local network. The structure of a fog node is shown in Fig. 5. IoT systems can generate data frequently and periodically. This data is stored in a compressed form in the Data Storage. A timestamp is used to determine which IoT device the data comes from and when. The *Communication Management* transfers the results it receives from the Data Distribution Service to the IoT devices through the CoAP protocol (*Yoon et al., 2019*). The *Security Management* carries out both security of fog and encryption of data to be sent to the cloud. The *Distributed*

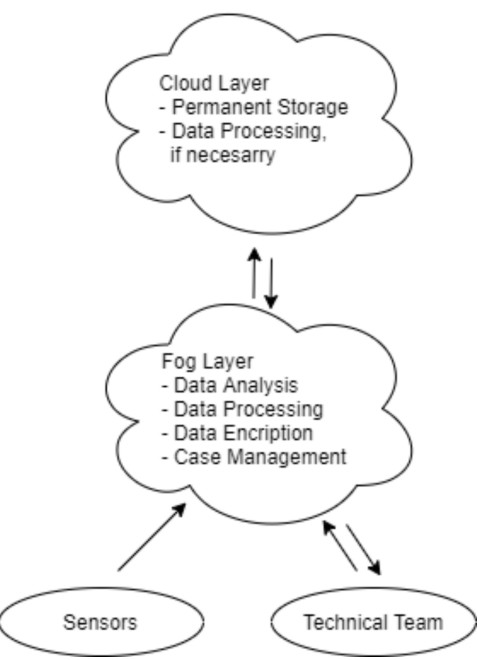

**Figure 6 Overview of tasks in the fog and cloud.**

*Management* provides management of data computing processes. The *Application Management* provides management of the applications used in the fog. The *SDN Controller* performs control and data integrity processes within the distributed blockchain-based architecture. Details of the blockchain based SDN structure used to verify the data and especially prevent flooding attacks are examined in the "Security of the Proposed Model".

In the proposed health and tactical analysis monitoring model, data is exchanged between end nodes, fog and cloud. The tasks that take place in the fog and cloud nodes are shown in Fig. 6.

Figure 6 shows the main tasks in the fog and cloud, and the path to raw data and result data. These collected data by sensors are sent to the fog layer. After processing in the fog layer, the results are either to the devices of technical team or to the cloud after being encrypted for permanent storage.

In Fig. 7, tasks in fog nodes correspond to data preprocessing and validation ($v_1$), data analysis and classification ($v_2$), computing of health data ($v_3$), computing tactical analysis data such as possible pass combinations and quality pass analysis thanks to Voronoi graph and Delaunay triangulation ($v_4$), encryption and decryption processes for data to and from the cloud ($v_5$), preparation of data to end nodes ($v_6$), preparation of data to the cloud ($v_7$). Each $e_{ij}$ stands for the relationships and data flow between the $i$ and $j$ tasks. Graph covers the processes of classifying, analyzing and sending the results to the end devices or the cloud after the data is perceived. The fog node structure is given in Fig. 5, it has management and storage areas that perform these operations. Apart from these, there are also fog servers that act as authentication and identity providers.

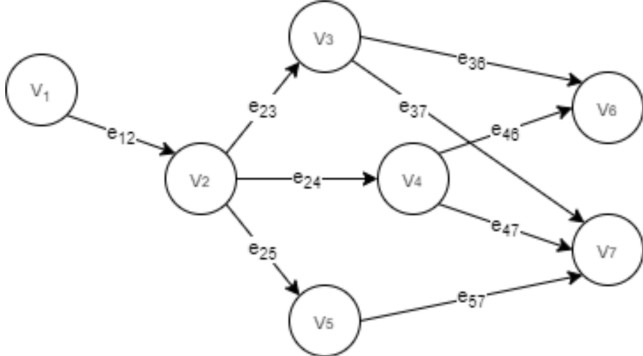

**Figure 7  Graph of tasks in fog nodes.**

IoT devices generate a lot of data for real-time processing with low latency tolerance. Therefore, task management algorithms are used in fog nodes. The proposed model uses a priority queue based on task management algorithm. The operations in Fig. 7 show the priority assignment step of this algorithm (*Choudhari, Moh & Moh, 2018*). Requests from the end devices arrive at the nearest fog node. There are three queues in the fog node: high ($Q_H$), medium ($Q_M$) and low ($Q_L$). Each request ($i$), is assigned to these queues according to the original priority level of the request ($SB_{CAT}$) and whether the request delay time (or total service time) ($delay_i^T$) is between the two thresholds ($T_1$ and $T_2$). This process is made with the priority assignment algorithm. In the fog nodes, the request delay time ($delay_i^T$) of a node request is equal to the difference of the deadline time given by the request and the arrival time to the fog node. The total time ($W_i$) for a request in the fog layer is equal to the sum of the request's wait time in the queue and the processing time of the request. Thus, in accordance with the Service Level Agreement (SLA) used in the algorithm, the request ($i$) must be $W_i < delay_i^T$ to supply the Quality of Service (QoS) requirement.

According to the Algorithm 1 (*Choudhari, Moh & Moh, 2018*); a request ($i$) is assigned the highest priority queue ($Q_H$) if the total service time ($delay_i^T$) of the request from the node is less than or equal to the estimated service time ($ST_{est}^i$) of the request ($i$). Otherwise, it is checked whether the total service time of the request is between the threshold values of $T_1$ and $T_2$. If it is within this range, the original priority level of the request ($SB_{CAT}$) is checked. Accordingly, the request is assigned to the queue $Q_H$ or $Q_M$ or $Q_L$. Finally, if the $delay_i^T$ value is greater than $T_2$, the requests are forced to be assigned to the queue that matches their original priorities. If the queues are full, they are assigned to a low level queue. Thus, higher priority and shorter processes are allowed to be executed.

According to the Algorithm 2, the management of fog nodes consists of two steps. In *Step 1*, Each request ($req_i$) is sent to the fog node ($node_{dt}$). $node_{dt}$ distributes tasks to other fog nodes. Each node estimates the total service time from the amount of unused resources. The estimated total service time of node k ($ST_{est}^k$) is sent to $node_{dt}$. This value is kept in ascending sorted list ($L_{node}$) in the $node_{dt}$. The list is constantly updated. Each $req_i$ is placed in queues according to the Algorithm 1. In *Step 2*, Requests in all queues

**Algorithm 1** Priority assignment algorithm.

$Priority(delay_i^T, ST_{est}^k)$

**if** $delay_i^T \leq ST_{est}^k$ **then**

    **return** (H)

**else if** $T_1 < delay_i^T \leq T_2$ **then**

    **if** $SB_{CAT} = 1$ **then**

        **return** (H)

    **else if** $SB_{CAT} = 2$ **then**

        **return** (M)

    **else if** $SB_{CAT} = 3$ **then**

        **return** (L)

**else if** $delay_i^T > T_2$ **then**

    **if** $SB_{CAT} = 1$ **then**

        **if** $Q_H$ *is not full* **then**

            **return** (H)

        **else**

            **return** (M)

    **else if** $SB_{CAT} = 2$ **then**

        **if** $Q_M$ *is not full* **then**

            **return** (M)

        **else**

            **return** (L)

    **else if** $SB_{CAT} = 3$ **then**

        **return** (L)

are processed in the order of $Q_H$, $Q_M$, $Q_L$. In the $L_{node}$ list in the node$_{dt}$, the request (req$_i$) is assigned to node$_k$ when the total service time ($ST_{est}^k$) of the node$_k$ is equal to or greater than the (delay$_i^T$) of the req$_i$. The loop is broken and the next request is handled. If the req$_i$ is not assigned to any fog node, the $L_{node}$ is reversed and looped again. If the total time (*sum*) is greater than or equal to delay$_i^T$, the req$_i$ is assigned to the group of nodes, and the loop is broken and the next request is handled. Otherwise, each value in $L_{node}$ until that time is summed. $L_{node}$(*first*) is equal to the $ST_{est}^{last}$ value before the list is reversed. If the req$_i$ is not assigned to the group of nodes, the fog layer cannot supply with the request. In this case, the queue with the request is based. If the request is in $Q_H$ or $Q_M$ queues, it is rejected. If the request is in the $Q_L$ queue, it can be sent to the cloud because the transmission of data to the cloud increases latency. The parameters and definitions of Algorithm 2 are shown in Table 1.

Equation 1 is used to determine the nodes that supply the reqi request and the limit for the nodes supplying the reqi.

---

**Algorithm 2** Management of fog nodes.

Step 1

*Each fog node sends the $ST_{est}^k$ (for node k) to $node_{dt}$*

*$node_{dt}$ assigns each $ST_{est}^k$ to the ordered $L_{node}$ list.*

**foreach** *$req_i$* **do**

    *$req_i$ is sent to $node_{dt}$*

    *$(Pr_i) = Priority(delay_i^T, ST_{est}^T)$*

    **if** *$Pr_i = H$* **then**

        *$Q_H(last) \leftarrow req_i$*

    **else if** *$Pr_i = M$* **then**

        *$Q_M(last) \leftarrow req_i$*

    **else if** *$Pr_i = L$* **then**

        *$Q_L(last) \leftarrow req_i$*

Step 2

*/* Task distribution to fog nodes according to $ST_{est}^k$ and $delay_i^T$ */*

**foreach** *$req_i$ in $Q_H, Q_M, Q_L$* **do**

    **for** *$L_{node}(k)$* **do**

        **if** *$ST_{est}^k \geq delay_i^T$* **then**

            *$node_k \leftarrow req_i$*

            **break**

    **if** *$req_i$ is not assigned to any fog node* **then**

        *reversedList $\leftarrow$ reverse($L_{node}(k)$) /* from high to low (processing capacity) */*

        *sum $\leftarrow L_{node}(first)$*

        **for** *reversedList* **do**

            **if** *sum $\geq delay_i^T$* **then**

                *$L_{node}(first), L_{node}(second), ..., L_{node}(k) \leftarrow req_i$*

                **break**

            **else**

                *sum $\leftarrow$ sum $+ L_{node}(next)$*

    **if** *$req_i$ is not assigned to any fog node group* **then**

        **if** *($req_i$ in $Q_H$) or ($req_i$ in $Q_M$)* **then**

            *reject($req_i$)*

        **else if** *$req_i$ in $Q_L$* **then**

            *cloud $\leftarrow req_i$*

---

$$L_{node}^i \leq \sum_{k=1}^{n} L_{node}^k, n \leq size(L_{node}), L_{node}^i \geq delay_i^T \tag{1}$$

The total capacity of the resources is determined according to Eq. (2) for each request ($L_{node}^U$, indicates the updated capacity, while $L_{node}^C$ indicates the current capacity).

---

**Table 1 Parameters for the proposed algorithm (Algorithm 2).**

| Parameters | Definitions |
|---|---|
| $Q_H$, $Q_M$, $Q_L$ | Priority queues |
| $req_i$ | Request |
| $delay_i^T$ | Delay cost for i request |
| $ST_{est}^T$ | Estimated total capacity of requests in queue |
| $ST_{est}^k$ | Estimated service capacity of node k |
| $node_{dt}$ | Distributor node |
| $L_{node}$ | Capacity list of ordered nodes |
| $L_{node}^k$ | k.Element of the node list |
| $node_k$ | k.Node |

$$L_{node}^U = L_{node}^C - L_{node}^i \qquad (2)$$

After each request is finished, the total capacity of resources is updated according to Eq. (3).

$$L_{node}^U = L_{node}^C + L_{node}^i \qquad (3)$$

In the Big-O complexity analysis for the Algorithm 1, $delay_i^T$ and $ST_{est}^i$ values are assumed to be calculated. Each request takes place in the complexity of $O(1)$. The total complexity for $n$ requests is $O(n)$. In the Big-O complexity analysis for the Algorithm 2, *Step 1* has the $O(n)$ complexity for $n$ requests because it uses the Algorithm 1. In *Step 2*, the number of requests in each queue to be processed is $n$, the number of fog nodes is $m$. The worst case is that the request is not assigned to any fog node. In this case, the worst complexity of *Step 2* is $n \cdot m + m + 6$. The complexity of *Step 1* and *Step 2* together is $n + n \cdot m + m + 6$. The complexity of the Algorithm 2 according to Big-O notation is $O(n \cdot m)$.

### Security of the proposed model

Perceived data is circulated a large number of nodes, such as processing in fog nodes, being sent back to end nodes, and stored in the cloud via the internet. Close-distance ZigBee networks have security mechanisms such as the AES-128 encryption method, and the renewal of keys in short periods (*Li, Jia & Xue, 2010*). In addition, authentication, data privacy and integrity must be ensured and data must be transmitted encrypted on the internet. Therefore, the implicit certificate-based FLAT method is used for authentication and data privacy in the proposed model (*Santos et al., 2020*). Data integrity is ensured and flooding attacks are detected with block chain based SDN controller. Secure communication between fog and cloud is performed by public key cryptography.

In the proposed model, the FLAT method is used for authentication between IoT sensors and fog nodes. Only the data of legal sensors are taken into fog nodes. If an attacking node successfully passes the FLAT authentication stage in any way, the data passes through the SDN controller. In this way, a warning is received in unreliable situations. The SDN controller does not warn in reliable situations. The security of the

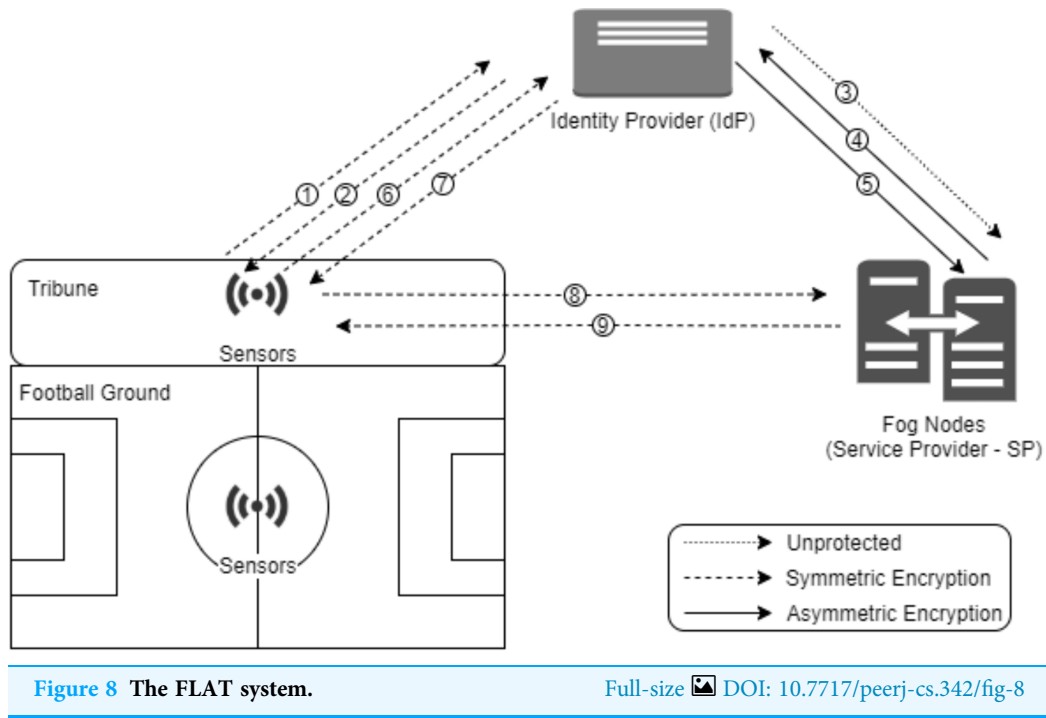

**Figure 8** The FLAT system.               

proposed model is provided in these two stages. However, in this article, the proposed task management algorithm is simulated.

### Data privacy and authentication

Federated Lightweight Authentication of Things is used in the proposed model for authentication and privacy. FLAT is the adaptation of Federated Identity Management (FIdM), the cloud authentication system, to fog computing (*Santos et al., 2020*). FIdM performs authentication and encryption processes at high cost. FIdM can be used to provide security between cloud and fog. However, authentication and encryption can be done with FLAT on resource-restricted devices. Therefore, FLAT protocol is used in the proposed model. The FLAT protocol consists of low computing client, high computing service and identity provider devices. Clients are resource-constrained sensor devices and user devices. The *Service Provider (SP)* is the fog server. The *Identity Provider (IdP)*, on the other hand, is a server that should work uninterruptedly in the fog network. For the FLAT protocol, the symmetric key must be preinstalled on the end devices. These keys are created with Physical Unclonable Functions (PUF) to be resistant to physical attacks. After the PUF output is produced, it is formatted and stored. Formatted PUF outputs are used to provide trust between end nodes and IdP. Trust between IdP and SP is provided by digital certificates given by a common certification provider.

According to Fig. 8 (*Santos et al., 2020*); when the resource-constrained client device wants to access the service, it requests a session key from the IdP (in 1). IdP transmits the session key to the client (in 2). IdP sends its certificate to SP; The SP sends its certificate to IdP (in 3, 4 and 5). After a secure communication channel has been created, IdP

sends the client key to the SP. As with the FIdM system, the client requests a confirmation from IdP and receives approval (in 6 and 7). The client requests service by presenting this confirmation package to SP (in 8). SP provides service if the confirmation package is correct (in 9). So the identity is verified and the SP starts the service.

Each sensor placed on players and the sports field must establish a secure session with fog nodes before sending data to fog nodes. For this reason, the FLAT authentication system is used in the model. The FLAT system offers a lightweight system for authenticating with resource constrained sensors. The communication of the sensors in the sports field, the identity provider and the service provider on the FLAT system is shown in Fig. 8. After this process, the task management process starts.

In the FLAT system, both the client and SP must rely on IdP. Communication between IdP and SP takes place using asymmetric key cryptography, while IdP-client and SP-client communications take place using symmetric key cryptography. Therefore, the FLAT protocol is more efficient for IoT systems using fog computing. To increase the security of the system against attacks (Soni, Pal & Islam, 2019) authentication method can be used between a user and a node that is a trusted authority.

The FLAT method uses Message Authentication Codes (MACs) and digital signature for authentication, symmetric and asymmetric encryption for confidentiality, MACs, digital signature and PUFs for integrity, authentication of two parties for availability, and the FIdM model that limits access to identity for privacy (Santos et al., 2020).

### Security with blockchain based SDN controller

Blockchain is a storage technology that serves as a registry, allowing the transactions performed in the system to be tracked by each node (Nofer et al., 2017). A blockchain consists of a chain of data packets (blocks) and is expanded with the new block added. Therefore, it represents a logbook. Fog nodes with blockchain based SDN controllers are used in the fog layer of the proposed model. The structure of the blockchain based SDN controller is shown in Fig. 9 (Sharma, Chen & Park, 2017). This structure is the SDN controller part in the fog node shown in Fig. 5.

Communication between end devices and fog nodes takes place with the ZigBee controller. The ZigBee controller is considered a gateway or routing switch for the SDN controller in the fog node. The SDN controller structure consists of three steps (Sharma, Chen & Park, 2017).

- *Step 1*: The *Packet Parser* carries out tracking and parsing operations to identify basic messages from arrived packets. Each packet must be captured and parsed to identify abnormal situations. The package parsing phase consists of four messages, *Features_Reply, Stats_Reply, Flow_Mod* and *Packet_In*. The attacker should capture a subset of these messages to change the network structure of the SDN controller. The packet parser dynamically tracks arrived packets to extract metadata.
- *Step 2*: The *Flow Topology Graph Builder* carries out the operations of parsing metadata properties set and network topology to analyze datasets from the packet parsing

phase and create a graph of the network flow topology. The logical and physical topologies of metadata for each flow are stored. The system distinguishes changes, malicious updates, and security strategy violations by looking at the graph of the stream topology created and byte/packet statistics transferred for each flow.

- *Step 3*: The *Verifier* confirms the metadata according to the management strategies defined by the analyst. It marks the known attacks in line with management strategies. It warns only when it recognizes an unreliable condition, not for every flow. The *Migration Agent* recognizes attacks like flooding and makes decisions after received alerts. These decisions are added to the reactive rules of the parser. In case of flooding attack, it sends the packets to the *Data Plane Cache* which is a temporary storage area, to prevent the controller from flooding and overloading. After updating the flow rule, the cached packets are processed.

Due to the blockchain-based SDN controller, data changes and flooding attacks are detected in the fog layer (*Sharma, Chen & Park, 2017*). Thus, data integrity is provided. The SDN controller method can be integrated into the proposed model to ensure data integrity during data transmission from sensors to fog nodes. In the proposed model, the arrived packages in Fig. 9 consist of raw data generated by the sensors. When this method is used, the data flow on the topology is examined before the task management starts in the fog nodes. Thus, data can be supported to reach the fog nodes safely.

## ANALYSIS OF THE PROPOSED MODEL

In this section, the proposed model is analyzed in terms of low latency, data accuracy, authentication and privacy, data integrity and accessibility, bandwidth savings.

### Low latency

Since fog computing is a locally operating mechanism, fog computing and IoT based health and tactical analysis monitoring model in sports have low latency. Thus, rapid intervention, which is very important in health, can be provided. It is also important that data processing and response transmissions are done quickly for technical issues. Research data from an artificial intelligence expert at the STATS analytics company, shows that it is important to take action quickly under a tactically negative situation: "You identify a specific scenario that tends to disrupt the opponent, giving you, say, a 30-s window where the opponent is disorganized" (*Burn-Murdoch, 2018*). Since the data is processed on fog servers without the Internet, the delay time in transmitting the response is low. In addition, response time and cost are further reduced with the fog task management algorithm based on process priorities (*Choudhari, Moh & Moh, 2018*). Data on which the response delay time is ignored is encrypted on the fog servers and processed or stored in the remote cloud server.

### Accuracy of perceived data

Correct positions must be taken from players and the ball to carry out tactical analyzes. Linear position sensors are used for this. By determining the positions of each node with

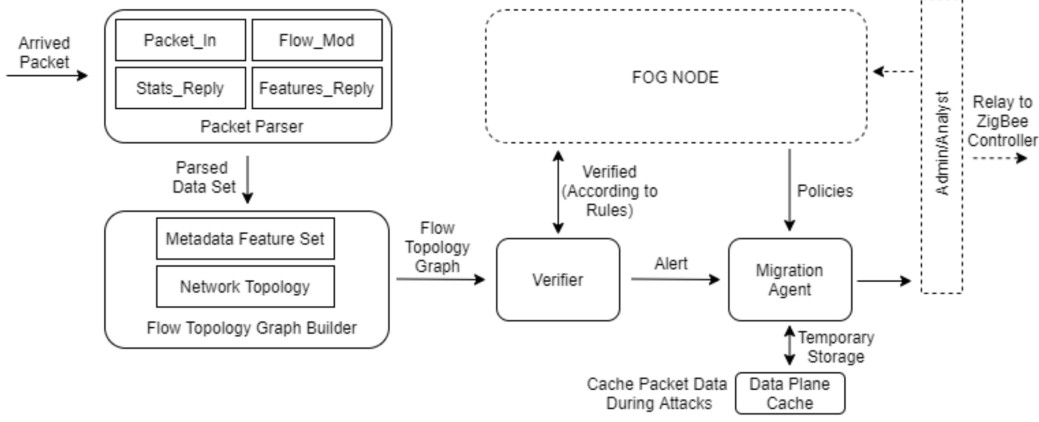

**Figure 9 The structure of the blockchain based SDN controller.**

linear position sensors, methods such as Voronoi graph, Delaunay triangle, dominant regions and shading are obtained. These methods help to make tactical analysis. ZigBee network is used for the devices to communicate with each other. The risk of data corruption is reduced due to the reasons such as the coverage of the ZigBee network within the field boundaries and using the AES-128 symmetric encryption structure in it. Therefore, the ZigBee protocol contributes to the accurate acquisition of data. Since the fog nodes are close to the end devices, the fog layer in our model reduces the risk of data corruption due to distance.

## Authentication and privacy

The proposed model uses the FLAT protocol for authentication and data privacy. FLAT is efficient for IoT applications with resource-constrained devices and using fog computing (*Santos et al., 2020*). In FIdM using cloud computing, the client is directed to IdP after communicating with SP first. However, in the FLAT protocol, the client directly contacts IdP to reduce this cost. In the certificate exchange between SP and IdP, the cost of asymmetric encryption is negligible, since both providers are servers with high computing power. Symmetric switches are assigned to formatted PUF output to clients with limited resources before the system starts processing. Therefore, client-SP and client-IdP communications are secured with symmetric encryption, which is lightweight. In the proposed model, asymmetric and symmetric encryptions are used for authentication and data security, and privacy is also provided.

According to the FLAT protocol, since the transported data is encrypted, data content cannot be accessed even if the data is captured in Man-In-The-Middle (MITM) attacks. Also, in order to prevent MITM, digital signatures are verified by exchanging certificates between IdP and SP before sending the shared key to SP. The client's identity is also verified by IdP using symmetric keys. The attacker cannot obtain the implicit certificate supplied by a legitimate certification authority and the private key of the fog node. Thus, impersonation attacks can be prevented because fog nodes use certificates and signatures.

## Data integrity and accessibility

In IoT applications using fog layer, the data perceived from the sensor devices are transmitted from the wireless environments to the fog nodes, so they may have vulnerabilities against attacks. Some security measures (such as encryption) are taken by the ZigBee network and the FLAT method, which is lightweight and secure, is used for authentication. FLAT also supports the protection of data integrity for fog nodes with methods such as digital signature. However, in order to protect the integrity of the data obtained from end nodes, blockchain based SDN controller structure is used in the fog layer. A flow topology graphic is created by parsing the arrived packets. Then, the data verification is performed in Verifier according to defined rules. This method is especially effective against flooding attacks. Packages are stored in the Data Plane Cache to prevent flooding and overloading of the SDN controller. Thus, the system continues to operate without loss of information and overflow. The cached packets are processed after high flow is over. Thus, the SDN controller can detect and warn attacks such as false topology, ARP poisoning and DDos (*Sharma, Chen & Park, 2017*). With blockchain's distributed data management, the cost of the system is reduced and its integrity and data security are ensured (*Yang, Cha & Song, 2018*).

## Bandwidth savings

Implicit certificate structure is used in certificate changes. The implicit certificate, also called the Elliptic Curve Qu-Vanstone (ECQV), is a public key certificate (*Santos et al., 2020*). Implicit certificates include ID, public key, and digital signature of the certification authority. Since these data are sent as a public key certificate, not separately, the certificate sizes are decreasing. Thus, bandwidth is significantly saved.

## SIMULATION OF THE PROPOSED MODEL

This section includes the detailed simulation of the proposed algorithm using the iFogSim program.

## Topology

The number of nodes in the fog layer and accordingly the topology structure can be changed. The simulation of the proposed algorithm, three fog nodes consisting of two levels are used in the fog layer. In Fig. 10, one of the three nodes is only responsible for communication with the cloud and performs storage tasks. The other two nodes carry out the requests and transmit the results to the actuators.

In Fig. 10, the node f-0 represents the numofGateways variable in the simulation, while the nodes f-1 and f-2 represent the numOfEndDevPerGateway variable in the simulation. The nodes s-0, s-1, s-2 and s-3 represent sensors in players and in the field of sports. The t-0 and t-1 nodes represent actuators such as technical team and club doctors. In the simulation, it is assumed that the data obtained from the sensors have successfully passed the security steps and are completely legal data. The simulation includes placing requests on the fog nodes and the proposed resource management algorithm.

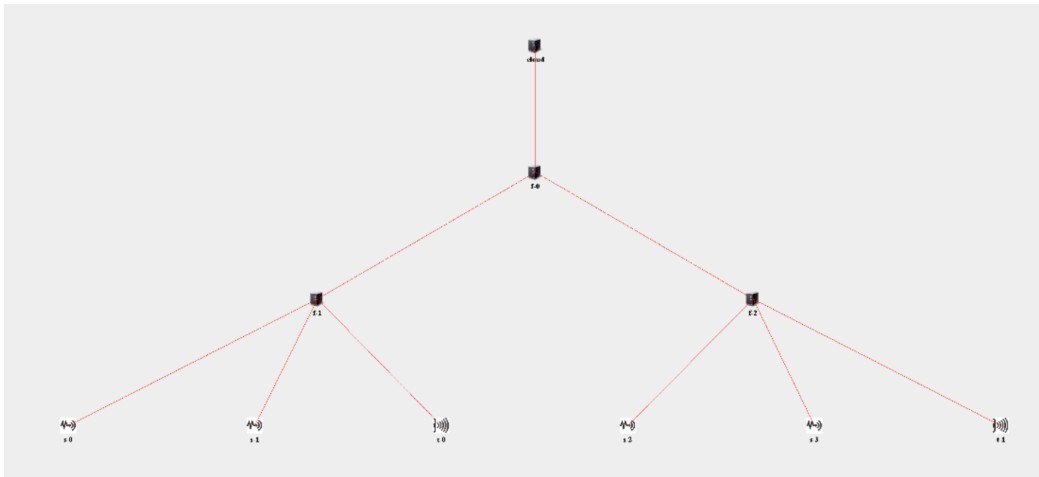

**Figure 10 The topology used in the simulation of the proposed algorithm.**

**Table 2 Features of fog nodes used in simulation.**

| Features | 1.level fog devices | 2.level fog devices |
| --- | --- | --- |
| Mips | 2,800 | 3,200 |
| Ram | 4,000 | 1,000 |
| upBw | 10,000 | 10,000 |
| downBw | 10,000 | 270 |
| Level | 1 | 2 |
| ratePerMips | 0 | 0 |
| busyPower | 107.339 | 87.53 |
| idlePower | 83.4333 | 82.44 |

## Features of fog nodes used in simulation

The features of the simulation application is taken as an example from (*Buyya & Srirama, 2019*). In the simulation, fog nodes are created with Linux operating system, x86 architecture and Xen as virtual machine manager. Additionally, the createFogDevice function is used to create a new fog node. Accordingly, The parameters of this function and the properties of the fog nodes are given in Table 2.

## Structure of the simulation application

Three modules are used in the application as "clientModule", "mainModule", "storageModule". It is assumed that it is placed in the "clientModule" end fog devices and the "storageModule" cloud device. "mainModule" is the starting module. The modules, the names given to the data exchange and the direction of the Tuples used in data transmission are shown in Fig. 11 (*Buyya & Srirama, 2019*).

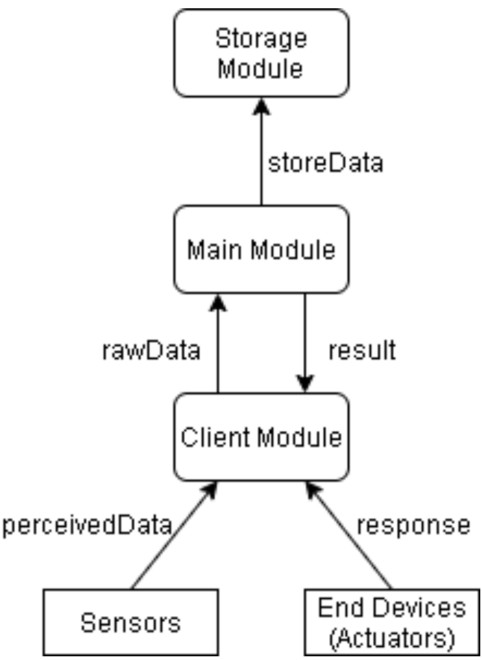

**Figure 11 Data exchange between modules in simulation.**

## The code structure of the simulation application

The number of fog nodes can be determined with the variables "numOfGateways" and "numOfEndDevPerGateway" in the "TestApplication.java" file where the main block of the simulation application is located in the iFogSim program. Small values (respectively 1 and 2) are given to these variables to show that the proposed algorithm is fully simulated. The object of the "MyModulePlacement.java" class is called for module placement in the main block in the simulation application. This class is designed according to the proposed algorithm. Since data flow cannot be done with the sensors in the simulation, the requests (mips and time values) are given to the system randomly through a list. These requests, which are assumed to be sorted according to their priorities, are assigned to the fog nodes sequentially according to the proposed algorithm. After each assignment, the list of fog nodes is sorted according to their unused capacity. In a loop, if the request can only be satisfied by one fog node, it is placed at that fog node and the loop breaks. In this case, the module placement process is carried out according to the code in Fig. 12.

If the request cannot be assigned to a single fog node, the list of fog nodes is scanned backwards and the capacities of the nodes are summed. If the appropriate capacity is reached, it is distributed across these fog nodes. In this case, the module placement process is carried out according to the code in Fig. 13.

If the request cannot be assigned to any fog node (single or distributed), the request is rejected because there is not enough capacity in the fog layer. This process is shown in the code in Fig. 14.

```
TestApplication.java      MyModulePlacement.java
166        if (requestMipsInfo.get(i)<=currentMips) {
167            currentMips = currentMips - requestMipsInfo.get(i);
168            sortedChildDeadline.put(key, currentMips);
169            System.out.println("The request  "+requestMipsInfo.get(i)+", is assigned to node with ID "+
170                    key+", for "+requestDeadlineInfo.get(i)+" units of time.");
171
172            sortedChildDeadline = sortedChildDeadline.entrySet()
173                    .stream()
174                    .sorted(Map.Entry.comparingByValue())
175                    .collect(Collectors.toMap(
176                    Map.Entry::getKey,
177                    Map.Entry::getValue,
178                    (oldValue, newValue) -> oldValue, LinkedHashMap::new));
179
180        System.out.println("Remaining sorted fog device capacity: "+sortedChildDeadline+"\n");
181
182        if(!getDeviceToModuleMap().containsKey(key)){
183            List<AppModule>placedModules = new ArrayList<AppModule>();
184            placedModules.add(appModule);
185            getDeviceToModuleMap().put(key, placedModules);
186        }
187        else{
188            List<AppModule>placedModules = getDeviceToModuleMap().get(key);
189            placedModules.add(appModule);
190            getDeviceToModuleMap().put(key, placedModules);
191        }
192
193        finishedRequestMipsInfo.add(requestMipsInfo.get(i));
194        finishedRequestDeadlineInfo.add(requestDeadlineInfo.get(i));
195        requestMipsInfo.remove(i);
196        requestDeadlineInfo.remove(i); i=0;
197        break;
```

**Figure 12  Module placement executed if the request can only be assigned to a fog node.**

```
TestApplication.java      MyModulePlacement.java
220    for(int k=sortedValues.size()-2; k>=0;k--){
221        AppModule appModule = getApplication().getModuleByName(moduleToPlace);
222        sum = sum + sortedValues.get(k);
223        usedKeys.add(sortedKeys.get(k));
224        usedValues.add(sortedValues.get(k));
225        if (request<=sum) {
226            int remain = request;
227            for (int l=0; l<usedKeys.size();l++) {
228                System.out.println("Part of the remaining request ("+remain+") is assigned to the node with ID "+
229                        usedKeys.get(l)+", for "+requestDeadlineInfo.get(i)+" units of time.");
230                if(remain-usedValues.get(l)>=0) {
231                    remain=remain-usedValues.get(l);
232                    usedValues.set(l,0);
233                } else {
234                    int x = usedValues.get(l)-remain;
235                    usedValues.set(l,x);
236                }
237                sortedChildDeadline.put(usedKeys.get(l), usedValues.get(l));
238                System.out.println("Remaining sorted fog device capacity: "+sortedChildDeadline+"\n");
239            }
240
```

**Figure 13  Module placement executed if the request could not be assigned to a single fog node.**

```
274            if(request>sum) {
275                AppModule appModule = getApplication().getModuleByName(moduleToPlace);
276                List<AppModule>placedModules = getDeviceToModuleMap().get(deviceParent);
277                placedModules.add(appModule);
278                getDeviceToModuleMap().put(deviceParent, placedModules);
279                System.out.println(" The request "+request+"is not assigned to any fog node.");
280                System.out.println("getDeviceToModuleMap"+getDeviceToModuleMap());
281
282            }
283
```

**Figure 14  Process executed when the request is not assigned to the fog layer.**

```
 Problems  @ Javadoc  Declaration  Console 
<terminated> TestApplication (1) [Java Application] C:\Program Files\Java\jre1.8.0_181\bin\javaw.exe  (23.Eyl.2020 15:45:51 – 15:45:52)
Starting Health and Tactical Analysis System...
Modulscloud and storageModule
Modulse-0-1 and clientModule
Modulse-0-0 and clientModule
Children: [5, 8]
Children deadline: {5=3200, 8=3200}
Requests: [1354, 564, 959, 1360, 1287, 1350, 1105]

Sorted fog device capacity{5=3200, 8=3200}

The request  1354, is assigned to node with ID 5, for 5.02158492476934 units of time.
Remaining sorted fog device capacity: {5=1846, 8=3200}

The request  959, is assigned to node with ID 5, for 5.719593011268303 units of time.
Remaining sorted fog device capacity: {5=887, 8=3200}

The request  1360, is assigned to node with ID 8, for 5.754760244818637 units of time.
Remaining sorted fog device capacity: {5=887, 8=1840}

The request  1287, is assigned to node with ID 8, for 3.2982788981888014 units of time.
Remaining sorted fog device capacity: {8=553, 5=887}

Part of the remaining request (564) is assigned to the node with ID 5, for 3.461690906820263 units of time.
Remaining sorted fog device capacity: {8=553, 5=323}

Part of the remaining request (564) is assigned to the node with ID 8, for 3.461690906820263 units of time.
Remaining sorted fog device capacity: {8=0, 5=323}

 The request 1350is not assigned to any fog node.
getDeviceToModuleMap{3=[org.fog.application.AppModule@65b3120a, org.fog.application.AppModule@6f539caf], 5=[o
 The request 1105is not assigned to any fog node.
getDeviceToModuleMap{3=[org.fog.application.AppModule@65b3120a, org.fog.application.AppModule@6f539caf, org.f
0.0 Submitted application test_app
==========================================
============== RESULTS ==================
==========================================
EXECUTION TIME : 647
==========================================
APPLICATION LOOP DELAYS
==========================================
[IoTSensor, clientModule, mainModule, clientModule, IoTActuator] ---> 11.443750000001904
==========================================
TUPLE CPU EXECUTION DELAY
==========================================
RawData ---> 3.850000000000364
ResultData ---> 0.1625000000003638
IoTSensor ---> 0.1312500000003638
StoreData ---> 0.19928571428681607
==========================================
cloud : Energy Consumed = 1.647329525633375E7
g-0 : Energy Consumed = 834332.9999999987
e-0-0 : Energy Consumed = 866557.4478125478
e-0-1 : Energy Consumed = 874740.100000026
Cost of execution in cloud = 4470494.540625081
Total network usage = 718.92
```

**Figure 15  Simulation results of the proposed algorithm.**

## Results of the simulation application

The simulation results according to the randomly generated (1,354, 564, 959, 1,360, 1,287, 1,350, 1,105) (mips) list of requests are shown in Fig. 15. At the same time, a randomly generated deadline is determined for each request.

In Fig. 15, numofGateways = 1, numOfEndDevPerGateway = 2 are selected. In the graphics in Fig. 16, the simulation is run by increasing the "numOfEndDevPerGateway" value. In the "Tuple CPU execution delay" heading in Fig. 15, the types and delay times of the Tuples transmitted between modules are shown. Source and target modules and types of requests are given in Table 3.

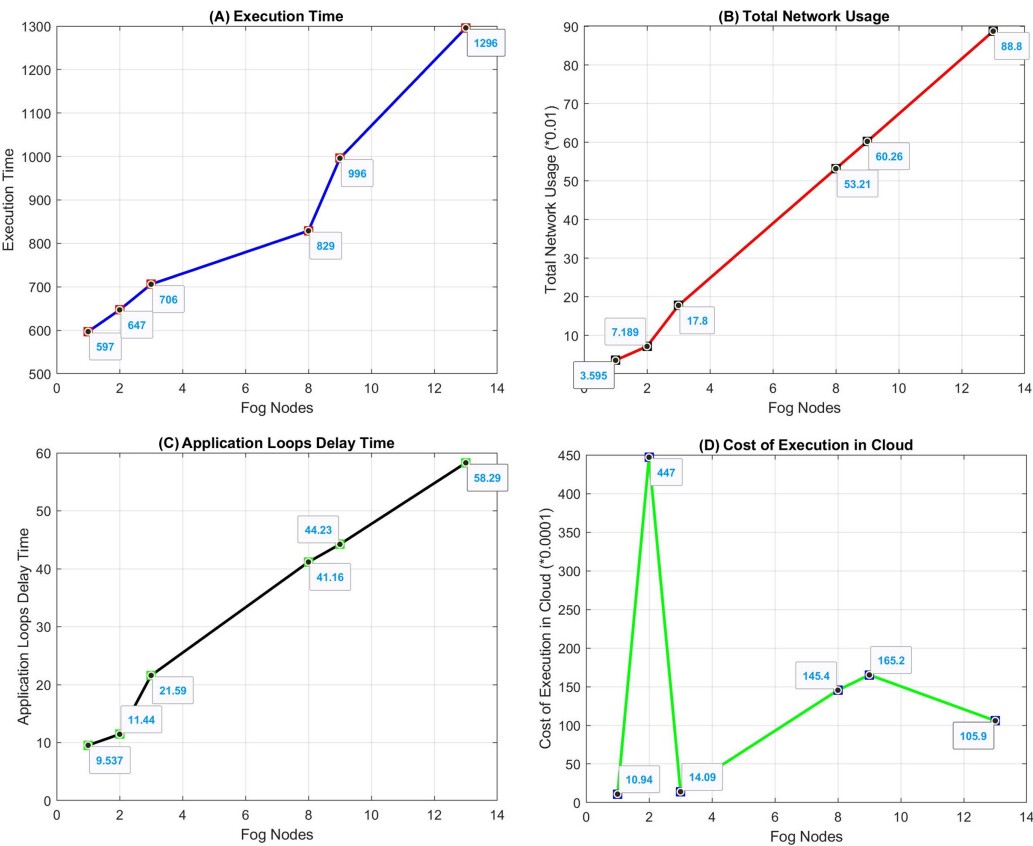

**Figure 16 Graphs of the simulation results of the proposed algorithm.** Graphs of the simulation results of the proposed algorithm ((A) Execution Time, (B) Total Network Usage, (C) Application Loops Delay Time, (D) Cost of Execution in Cloud).

**Table 3 The flow and types of tuples between modules.**

| Source | Destination | Tuple Type | Direction |
|---|---|---|---|
| IoTSensor | clientModule | IoTSensor | UP |
| clientModule | mainModule | RawData | UP |
| mainModule | storageModule | StoreData | UP |
| mainModule | clientModule | ResultData | DOWN |
| clientModule | IoTActuator | Response | DOWN |

The area in the node or nodes where the request is placed during the deadline period is reserved for that request. In the simulation, assuming that the placed requests have not yet produced results, the other requests in the queue continue to be placed in fog nodes. In order to show the result of each phase, the number of fog nodes was chosen as 2. For results that occur with more fog nodes, it can be re-run by changing the values of "numOfGateways" and "numOfEndDevPerGateway" in the main block of the simulation application. In Fig. 15, the "RESULTS" part, produced as a standard by the iFogSim program, shows the time spent for the operation of the system, the energy spent and the general cost.

**Table 4 Comparison of the proposed algorithm with the related work.**

| Features | Buyya & Srirama (2019) | The Proposed Algorithm |
|---|---|---|
| Execution Time | 124 | 647 |
| Total Network Usage | 809.3333333334 | 718.92 |
| Application Loop Delays Time | 17.71321428571426 | 11.443750000001904 |
| Cost of Execution in Cloud | 3,407.7379464284923 | 4,470,494.540625081 |
| Tuple Cpu Execution Delay (RawData) | 4.2317142857143 | 3.8500000000000364 |
| Tuple Cpu Execution Delay (ResultData) | 0.19075000000002545 | 0.1625000000003638 |
| Tuple Cpu Execution Delay (IoTSensor) | 0.13125000000002274 | 0.1312500000003638 |
| Tuple Cpu Execution Delay (StoreData) | 0.12704761904762257 | 0.19928571428681607 |

The graph of Execution Time (a), Total Network Usage (b), Application Loop Delays Time (c), Cost of Execution in Cloud (d) values from the simulation results obtained by changing the number of fog nodes is shown in Fig. 16. As the number of nodes increases, execution time and application loop delay increase. When the number of nodes is 4, high latency occurs in the RawData Tuple. Therefore, execution time and application loop delay are much higher. While the total network usage increases with the number of nodes, it decreases when there are 4 nodes. The high delay of the RawData Tuple does not affect network activities. In addition, the total network usage may vary depending on the request costs. Cloud usage cost is highest when the number of nodes is 2. This situation varies according to the cost values of the requests. Address for simulation codes: github.com/akarakaya/SimApp.

## Comparison of the proposed model

The proposed study is compared with Buyya & Srirama (2019), which shares the simulation codes openly. Both studies are compared using numOfGateways = 1 and numOfEndDevPerGateway = 2 parameters. In addition, the fog nodes with the features in Table 2 are used. The comparison results are shown in Table 4.

According to the comparison results, data transmission delay times between modules and network usage are lower in the proposed model. The cloud node costs are higher than the fog node costs. Additional protocols and remote server are used in communication with the cloud nodes. This indicates that fog computing causes lower latency than cloud computing. Therefore, fog computing is used in the proposed algorithm. In Table 4, the values of the proposed model are taken from the result in Fig. 15. If every request in the simulation is supplied by the fog layer in the proposed algorithm (if the situation in Fig. 14 does not occur), the values of the proposed model in Table 4 are much lower. However, this also depends on the size of the requests and the capacity of the fog nodes.

## CHALLENGES IN WEARABLE TECHNOLOGIES AND FUTURE WORKS

This section includes challenges in wearable technologies and some future study suggestions. When wearable device technology is combined with fog computing, this

integrated structure can be a solution for applications requiring low latency (*Fiandrino et al., 2019*). However, today there are a lot of challenges with wearable technologies. Therefore, the issue of wearable technologies was not included in this article.

Challenges in wearable technologies;

- Devices may not be suitable for every user's body. For this reason, the most appropriate and ergonomic product is tried to be developed according to age groups (*Hänsel et al., 2015*).

- With the interdisciplinary study between psychologists and engineers for new healthcare device designs, the product is addressed in terms of both technological and behavioral change. Since wearable health devices self-diagnosis without medical knowledge may mislead the user, this situation should be handled in the developed wearable products. The future of wearable applications should support not only the development of new analysis algorithms for health data, but also the feedback of users for behavior change (*Hänsel et al., 2015*).

- The size, flexibility and operating requirements of existing chemical sensors used for health perception are difficult to use in applications as they are not compatible with wearable technology (*Bandodkar, Jeerapan & Wang, 2016*). Existing wearable devices do not meet the requirements due to their low power supplies, low energy density and slow charging. There are also major challenges in processing and securing the large data produced by wearable sensors. New generation cryptographic algorithms need to be developed to ensure data security and user privacy (*Bandodkar, Jeerapan & Wang, 2016*).

Some future study suggestions:

- Wearable technologies have not yet reached the desired levels in terms of size, capacity, energy consumption, watertightness, high data processing, and security. For this reason, wearable technologies that are suitable for players and spectators, provide accurate data and are not affected by external factors such as sweat can be developed for sports applications.

- Lightweight and strong encryption and authentication methods can be developed for IoT and fog computing applications.

- Consensus algorithms that are efficient on resource-constrained devices can be developed for blockchain-based IoT applications.

- Lightweight and safe IoT models can be developed for different areas similar to the health and tactical analysis monitoring model in the sports we proposed.

- Although the elliptic curve encryption system uses smaller keys, it provides similar security with RSA. It is based on the discrete logarithm problem. The ECC-based authentication models have been designed to prevent many attacks (*Islam & Biswas, 2013*). Similar to ECQV implicit certificate method, post-quantum cryptographic protocols can be developed.

- Communication of IoT nodes, gateways, fog nodes, cloud nodes should be secure. All IoT systems are affected by post-quantum security due to the needs of IoT systems such as privacy, authentication, integrity (*Fernández-Caramés, 2019*). Therefore, quantum-resistant schemes based on post-quantum cryptographic algorithm can be developed for IoT and cloud systems.
- Authentication protocol based on post-quantum cryptography has been developed. This protocol, called LB-2PAKA, was analyzed in a random oracle model to measure provable security and to estimate the breaching time (*Islam, 2020*) Protocols based on post-quantum cryptography can be developed for IoT applications.
- In *Akleylek et al. (2019)*, post-quantum identification scheme was proposed in IoT applications. Various polar forms of multivariate quadratic and cubic polynomial systems was used in this scheme. In *Akleylek & Seyhan (2020)*, an authenticated key exchange approach based on the Bi-GISIS problem was proposed for post-quantum secure. Similarly, identification and authenticated key exchange methods based on post-quantum cryptography can be developed for IoT applications.

## CONCLUSION

In this study, a light and safe fog-based IoT model is proposed in which the result obtained by analyzing the health and tactical data in sports is reported to the technical team and club doctors. In the proposed model, it is ensured that the responses to the end nodes are transmitted with low latency, and the data is processed and stored without the Internet. Urgent processes are given high priority with the priority queue method in the fog nodes. In this article, a resource management algorithm using priority queue method is recommended in the fog nodes. The algorithm is simulated using the iFogSim simulation tool. Simulation results and comparisons with a similar study show that resource allocation and data processing in the fog nodes are performed with low latency. In similar studies in sports, IoT applications are generally designed using only cloud computing. Only health monitoring systems are covered. However, in the proposed model, fog computing is used for data processing and storage. Thus, a higher performance architecture is created in terms of both security and low latency. In the proposed model, not only health monitoring status, but also tactical analyzes in the field are discussed. Thus, the technical team and the club doctors are warned in a short time against possible negative results. For authentication, FLAT protocol, a lightweight authentication protocol, is used in resource-restricted devices. Thanks to the implicit certificates in the FLAT protocol, significant savings in bandwidth can be achieved. In addition to authentication, this protocol provides data privacy. It also helps with data integrity in the fog layer. With the blockchain-based SDN controller structure, some attacks are detected and data integrity is provided. In the event of a flooding attack, data packets are stored in the cache and processed later to prevent flooding and overloading of the fog node. A lightweight and safe model is proposed for monitoring health and tactical analysis in sports. The proposed algorithm has been simulated and comparison results are given.

In the future; studies on artificial intelligence methods that use the proposed model and perform health and tactical analysis in fog nodes are planned.

### Funding
The authors received no funding for this work.

### Competing Interests
Sedat Akleylek is an Academic Editor for PeerJ.

### Author Contributions
- Aykut Karakaya conceived and designed the experiments, performed the experiments, analyzed the data, performed the computation work, prepared figures and/or tables, authored or reviewed drafts of the paper, and approved the final draft.
- Sedat Akleylek conceived and designed the experiments, performed the experiments, analyzed the data, performed the computation work, prepared figures and/or tables, authored or reviewed drafts of the paper, and approved the final draft.

### Data Availability
The simulation application codes are available at GitHub:

https://github.com/akarakaya/SimApp.

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
