# Peer review of "A novel IoT-based health and tactical analysis model with fog computing"

_PeerJ Computer Science, doi:10.7717/peerj-cs.342_

## Round 0.1 · original submission · Major Revisions

· Academic Editor

Major Revisions

Dear Authors,

Please revise and resubmit the manuscript based on the reviewers' comments.

·

Basic reporting

- This paper proposes an analysis model as a paper that applies Fog Computing-based IoT to the sports field.

- It is necessary to organize the research part of the existing research into a separate chapter. In addition, it is necessary to add a comparative evaluation part of this study and the existing one.

Experimental design

- As a result of analysis through simulation, it is necessary to verify experimentally how the analysis is excellent in terms of efficiency and security with existing research results.

- In addition to the logical argument that the analysis of data collected by the sensor at the Fog layer provides low-latency by being feedback directly to the device layer, it is also necessary to add simulation results to support this.

- We need a detailed explanation of the analysis model, which is the main topic of this paper. For example, what AI algorithms are used to provide the results of the analysis to the players and speculators.

Validity of the findings

- It is necessary to organize the evaluation contents by comparing and analyzing the existing relevant research results.

- Further analysis of providing security will also need to be added by applying FLAT. In particular, analysis is required from the perspective of security parameters in the field of sports.

- And as a solution to define and solve security issues that need to be solved in the sports field, an explanation is needed from the integrated perspective of security provided by Blockchain-based SDN controller and FLAT.

Reviewer 2 ·

Basic reporting

The paper proposes "A Novel IoT-Based Health and Tactical Analysis Model with Fog Computing". The idea is novel, however, the paper has a lot of technical mistakes that needs to be revised. The paper needs an extensive proofreading and professional English should be used. The paper has no mathematical formulation that can verify the effectiveness of the proposed algorithm. The author has used the SDN approach, however, there are no details of how the algorithm interacts with the controller. The results are not well drawn and written.

Experimental design

The experimental design needs more explanation. Which simulator is used for emulating the SDN.

Validity of the findings

The graphs and results are not professionally written and well explained. There is not flow and link between the algorithm and result section.

Additional comments

The author paper proposes "A Novel IoT-Based Health and Tactical Analysis Model with Fog Computing". The idea is novel, however, the paper has a lot of technical mistakes that needs to be revised.
1. The paper needs an extensive proofreading and professional English should be used.
2. The paper has no mathematical formulation that can verify the effectiveness of the proposed algorithm. 3. The author has used the SDN approach, however, there are no details of how the algorithm interacts with the controller.
4.The results are not well drawn and written.
5. The algorithm is not well defined, how will be the delay reduced using Fog computing approach.
6. What is role of sdn controller with Fog computing, how will fog computing receive the optimal paths.
7. Is fog computing intelligent or the SDN controller.

---

## Round 0.2 · Minor Revisions

· Academic Editor

Minor Revisions

More explanation is required regarding security process from the point of view of SDN controller.

·

Basic reporting

no comment

Experimental design

no comment

Validity of the findings

Please explain the security process in more detail from the perspective of FLAT and SDN controller.

Additional comments

Please review the sentences more about the overall logic of the paper.

Reviewer 2 ·

Basic reporting

The revision of the paper has been well addressed according to my comments. It can be acceptable.

Experimental design

The simulation details have been added and graphs have been well explained according to my concern.

Validity of the findings

Findings are validated.

Additional comments

The comments have been well addressed and the paper can be accepted.

---

## Round 0.3 · accepted · Accept

· Academic Editor

Accept

The manuscript has been improved as per comments of the reviewers.

·

Basic reporting

Good

Experimental design

Good

Validity of the findings

It can be acceptable.

Additional comments

It can be acceptable.